# Common Myelin Regulatory Factor Gene Variants Predisposing to Excellence in Sports

**DOI:** 10.3390/genes12020262

**Published:** 2021-02-11

**Authors:** Beata Pepłońska, Agata Piestrzynska-Kajtoch, Kinga Humińska-Lisowska, Jakub Grzegorz Adamczyk, Marcin Siewierski, Artur Gurgul, Agnieszka Fornal, Monika Michałowska-Sawczyn, Cezary Żekanowski, Paweł Cięszczyk, Mariusz Berdyński

**Affiliations:** 1Department of Neurodegenerative Disorders, Mossakowski Medical Research Institute, Polish Academy of Sciences, 02-106 Warsaw, Poland; beatap@imdik.pan.pl (B.P.); c.zekanowski@imdik.pan.pl (C.Ż.); 2Laboratory of Molecular Genetics, Department of Animal Molecular Biology, National Research Institute of Animal Production, 32-083 Balice, Poland; agata.kajtoch@izoo.krakow.pl (A.P.-K.); artur.gurgul@urk.edu.pl (A.G.); agnieszka.fornal@izoo.krakow.pl (A.F.); 3Faculty of Physical Education, Gdansk University of Physical Education and Sport, 80-336 Gdansk, Poland; kinga.huminska-lisowska@awf.gda.pl (K.H.-L.); monikamichalowska@op.pl (M.M.-S.); cieszczyk@poczta.onet.pl (P.C.); 4Department of Theory of Sport, Józef Piłsudski University of Physical Education, 00-968 Warsaw, Poland; jakub.adamczyk@awf.edu.pl (J.G.A.); marcin@funactive.pl (M.S.); 5University Centre of Veterinary Medicine UJ-UR, University of Agriculture in Krakow, 30-059 Krakow, Poland

**Keywords:** athletic performance, single nucleotide polymorphism, endurance sports, power sports, combat sports, myelination

## Abstract

In all sport disciplines, excellent coordination of movements is crucial for achieving mastery. The ability to learn new motor skills quickly and effectively is dependent on efficient myelination which varies between individuals. It has been suggested that these differences may play a role in athletic performance. The process of myelination is under transcriptional control by Myelin Regulatory Factor (MYRF) as well as other transcription factors (SOX10 and OLIG2). We analyze a panel of 28 single nucleotide polymorphisms (SNPs) located within the frequencies of common variants of *MYRF*, *SOX10* and *OLIG2* genes in professional athletes compared to non-athletes. No significant differences were detected after correction for multiple testing by false discovery rate (FDR) for any of the models tested. However, some deviations from the expected distribution was found for seven SNPs (rs174528, rs139884, rs149435516 and rs2238001, rs7943728, rs61747222, and rs198459). The MYRF alleles rs7943728 and rs61747222 showed a correlation with the level of sport achievement among the athletes. Even though the athletes did not differ from the non-athlete controls in the distribution of most SNPs analyzed, some interesting differences of several variants were noted. Presented results indicate that genetic variants of *MYRF* and *SOX10* could be genetic factors weakly predisposing for successful athletic performance.

## 1. Introduction

Myelinated white matter plays a central role in brain messaging, and likely mediates processing speed, which is an important contributor to working memory performance and fluid intelligence [1]. Working memory allows athletes to integrate and manage multiple proprioceptive-related variables, such as judging distance, appropriate force, and joint alignment [2], which is of vital importance for keeping attention and for decision-making processes. Its efficiency varies between individuals and it has been suggested that these differences may play a role in athletic performance [3]. Most sports require numerous higher-order cognitive abilities and are performed under extreme stress where human limits are being continually challenged and extended. In such situations, even minute differences in the performance affect the classification in a competition [4].

The condition of myelin sheaths influences the speed of saltatory conduction of nerve impulses and therefore efficiency of the nervous systems. In the central nervous system (CNS) myelin is synthetized by oligodendrocytes from proliferating precursors [5]. Developmental myelination continues at least into the fourth decade of life. This process appears to be adaptive and dynamic as both the absolute level of myelination and its rate vary throughout life [6,7]. 

The process of myelination is under direct transcriptional control by Myelin Regulatory Factor (MYRF, MRF) as well as other transcription factors, e.g., SOX10 and OLIG2 [8]. MYRF promotes myelin gene expression directly, as well as increases the expression of genes that modulate myelination. It is both indispensable for myelination during CNS development and for myelin maintenance in the adult [9]. MYRF interacts physically and functionally with SOX10, which is a direct regulator of myelin gene expression during oligodendrocyte differentiation [10,11]. Gain-of-function and loss-of-function genetic studies suggest that the *OLIG2* gene is critical for mediating oligodendrocyte development [12].

Using a transgenic mouse model, the activation of Myrf has been shown to be required for myelination during adulthood and the myelination to be required for motor skill learning [13]. Concomitantly, motor learning increased oligodendrocyte production. The same has been observed in human subjects trained in complex sensorimotor tasks (playing the piano, juggling) [14,15].

In all sport disciplines, excellent coordination of movements is crucial for achieving mastery. Athletic movements combine gross motor skills (involving a large muscle mass for typically large joint movements) and fine motor skills (control small ranges of high-precision movements). These very complicated and accurate motor patterns are learned through repetition during training. The ability to learn new motor skills quickly and effectively is dependent on efficient myelination. An extraordinary capability of fast modification and memorization especially of minor changes in movement sequence seems to be essential for achieving outstanding sports performance. Studies using mouse Alzheimer’s disease models have indicated that running exercise influence positively learning and spatial memory performance [16]. This suggests that sport exercise and myelin plasticity in the CNS are interrelated forming a positive feedback loop. 

Given the above observations, we decided to analyze the frequencies of common variants of *MYRF*, *SOX10* and *OLIG2* genes in professional athletes compared to non-athletes from the Polish population. According to our knowledge, it is the first association study of selected genetic variants of the aforementioned genes with outstanding sports achievements. We hypothesized that variants of the *MYRF* gene and of other genes encoding transcription factors important for myelination can influence the performance in sports in which a coordinated activity of numerous muscles, as well as precise movements, are important. 

## 2. Materials and Methods

The study participants were enrolled independently, out of two research centers, to two groups (study and validation), each comprising elite athletes and non-athlete controls. All participants were unrelated and of Polish origin. Their geographical distribution did not differ systematically between the two groups. 

The study group comprised 1597 participants (642 athletes and 724 controls, Table 1), and the validation group comprised 538 participants (257 and 281, respectively, Table 2). The elite athletes were recruited from various sports and the main inclusion criterion was outstanding performance at an international or national level, as described previously [17]. The inclusion criteria were the same for both groups. 

Written consent was obtained from all the participants according to the Declaration of Helsinki (*BMJ*
**1991**, *302*, 1194). The study was approved by the Ethics Committee of the Józef Piłsudski University of Physical Education in Warsaw in compliance with national legislation and the Code of Ethical Principles for Medical Research Involving Human Subjects of the World Medical Association.

Blood or saliva samples were obtained from participants between 2010 and 2016. DNA was extracted from peripheral blood lymphocytes using a standard salting-out method or from saliva using Orange DNA Self-Collection Kit and Prep IT L2P Purification Kit (DNA Genotek Inc., Ottawa, Canada). SNPs were genotyped using real-time PCR with a predesigned TaqMan OpenArray Genotyping System (Appendix A) provided by Applied Biosystems (by ThermoFisher Scientific, Waltham, MA, USA). The genotyping of the study group was performed following the manufacturer’s protocol on a QuantStudioTM 12K Flex Real-time PCR System with Open Array block and OpenArray™ AccuFill™ System (by ThermoFisher Scientific, Waltham, MA, USA). Validation group was genotyped on CFX Connect Real-Time PCR Detection System (Bio-Rad, Hercules, CA, USA) using TaqPath™ ProAmp™ Master Mix for material amplification according to the manufacturer protocol. The data were analyzed using the TaqMan^®^ Genotyper Software version 1.3 (Applied Biosystems by ThermoFisher Scientific, Waltham, MA, USA) and CFX Maestro Software version 1.1 (Bio-Rad, Hercules, CA, USA).

A panel of 28 SNPs located within the *MYRF*, *SOX10* and *OLIG2* genes were genotyped (Table 3). The presence of SNPs, as well as minor allele frequency (MAF) in the European populations (according to the 1000Genomes database, ExAC, and in-home unpublished WGS results) were the criteria for SNPs selection. The SNPs were scattered in the intronic and exonic sequences of the genes (synonymous and missense) to cover uniformly the genes. 

Results obtained for the study group were used to select eight SNPs for a further analysis in the validation group using TaqMan SNP genotyping Assays (ThermoFisher Scientific, Waltham, MA, USA) (Table 3). 

The quality of the analysis of each SNP was assessed according to the criteria proposed by Roberts et al. 2009 [18]. OpenArray fluorescence distribution plots for all samples were analyzed and variants with unsatisfactory separation were excluded (rs143193141 and rs142113652). The assays for these variants also had low call rates—93.6% and 86.7%, respectively. All other assays showed a call rate of about 95% and higher. Additionally, 273 samples (17.1% of the total number of samples) were run in duplicates, for various reasons, e.g., undetermined calls, lack of amplification, as technical replicates. The accuracy of the genotyping assessed by comparing calls from technical replicates was >99%.

The SNPs genotypes were checked for quality by evaluating the percentage of missing genotypes and deviation from Hardy–Weinberg equilibrium (HWE). Differences in the distribution of alleles and genotypes between various groups (see further) were analyzed assuming dominant and recessive models, using the Fisher exact test to deal with the low-frequency variants. The applied statistics included, apart from the standard allelic test, a Cochran–Armitage trend test, genotypic (2 df), dominant gene action (1df), and recessive gene action (1df) tests. The Cochran–Armitage trend (CA Trend) test for allele distribution was additionally used, as it does not assume Hardy–Weinberg equilibrium, where the individual, not the allele, is the unit of analysis. All pairwise comparisons possible were versus controls and among various sports types. In each comparison, monomorphic markers were excluded and *p*-values obtained were corrected for multiple testing (for each test type separately) using false discovery rate (FDR) methodology [19]. The correlation of SNPs with the degree of sport achievements (non-elite, elite and high elite) in the whole cohort or in sport-type subgroups was additionally tested using linear regression. Estimated odds ratios for the minor allele, as well as lower and upper bounds of 95% confidence interval for odds ratios, were also calculated. A pairwise linkage disequilibrium (LD) r2 coefficient (the square of the correlation coefficient between two indicator variables) among SNPs located on the same chromosome was used to evaluate possible SNPs linkage. All calculations were performed using Plink Whole-genome association analysis toolset software [20]. 

## 3. Results

### 3.1. Study Group Analysis

All analyzed SNPs but one (rs2238001) were in Hardy–Weinberg equilibrium (HWE). It was not in HWE (*p* < 0.05) in the control group and for all samples, but it met the HWE criteria in the athletes taken together or in individual sport-type subgroup. Missing genotypes were observed for six SNPs. The average percentage of missing calls for these SNPs did not exceed 0.17%. Most of the SNPs showed low polymorphism across the cohort with an average MAF of 0.075 (±0.13) (Appendix A). Eight SNPs were monomorphic and were therefore excluded from the analysis. To control additionally for marker polymorphism, filtration of monomorphic markers was redone for each between-group comparison, leaving for statistical analysis between 13 and 18 polymorphic markers. Of the whole SNPs panel tested, five low-frequency SNPs were detected only in the controls (rs2286008, rs139124174, rs143799782, rs143059056, and rs201384645). Two SNPs (rs198459 and rs149803) showed a high level of linkage disequilibrium in the whole cohort, with r2 = 0.51. Another pair of SNPs, rs174528 and rs7943728, showed an intermediate r2 value of 0.29, whereas all the remaining SNPs showed low levels of LD with a mean r2 of 0.008 (±0.022).

No significant differences in the distribution of alleles and genotypes were detected for any of the models tested and any pair-wise comparison after correction for multiple testing by FDR. However, some minor deviations from the expected distribution on a point-wise level were found for four SNPs (rs174528, rs139884, rs149435516 and rs2238001). The difference for SNP rs139884 was found nearly significant (*p* < 0.08) or significant (*p* < 0.05) (Appendix A), in the recessive model, in three comparisons involving the endurance group against combat sports, power sports, and controls. These differences resulted mainly from a lower frequency of the AA genotype (at the expense of two others) in the endurance group than in the others. SNP rs174528 showed some trend in the distribution of alleles between the combat and endurance groups. It was associated with a higher frequency of the minor C allele in the combat group (OR = 1.27, 95%CI = 0.965–1.671, *p* = 0.086). SNP rs149435516 showed a slightly lower frequency of the minor A allele in the endurance group vs. controls (OR = 0.5972; 95%CI = 0.3404–1.048, *p* = 0.08). SNP rs2238001 showed some deviation in genotype distribution between power and control groups, primarily assuming recessive model (*p* = 0.09), due to a higher frequency of the CC genotype in the power group. Appendix A shows data for all comparisons.

### 3.2. Validation Group Analysis

Eight SNPs selected as most promising basing on the results obtained for the study group were analyzed in the validation group. All these SNPs were in Hardy–Weinberg equilibrium. No significant differences in the distribution of alleles and genotypes were detected for any of the models tested and any pair-wise comparison after correction for multiple testing by FDR (Appendix A). As in the study group, three SNPs (rs7943728, rs61747222, and rs198459) showed statistically significant (*p* = 0.05) or nearly significant (0.08) differences in the distribution between cohorts. These SNPs were rare in both the study and the validation groups. The *MYRF* alleles rs7943728 and rs61747222 associated with the performance level (*p* = 0.06 and *p* = 0.04, respectively), with the minor allele A of rs7943728 significantly less frequent and rs61747222 more frequent in the Olympic/World-class athletes than in national-level athletes (Appendix A). A similar association was observed among endurance athletes (rs7943728, *p* = 0.07) and combat athletes (rs61747222 *p* = 0.08) (Appendix A). In addition, the minor allele of SNP rs7943728 was less frequent in the endurance subgroup than in the control group (genotypic test (*p* = 0.68) and dominant model (0.76). Similarly, the minor allele A of rs61747222 was less frequent among the combat athletes than in controls in all models. Moreover, the distribution of rs198459 alleles was significantly different between the combat athletes subgroup and the endurance subgroup (Appendix A). 

Comparison of genotypes revealed no statistically significant differences between the study group and validation groups (Appendix A). We did not perform statistical analysis for the study and validation groups combined. The statistical analyses of the study group and validation group were independent experiments performed using different genotyping techniques. A validation group experiment was performed after the study group experiments to independently confirm the obtained results. 

Samples obtained from 42 athletes were enrolled independently by two research centers. Therefore samples were genotyped both in study and validation groups. The genotypes of duplicated samples were concordant. The duplicated samples (*n* = 42) were not included in the validation group and statistically analyzed only in the study group

## 4. Discussion

Transcription factors SOX10, OLIG2, and MYRF play a pivotal role in the regulation of oligodendrocyte differentiation and significantly affect the functioning of the nervous system. Myelination in the human CNS begins 1 to 2 months prior to birth and lasts through the 3rd–4th decade of life. It seems plausible that variants of the genes encoding these transcription factors could affect the process of myelination by modulating the expression of numerous relevant genes, and as a result affect athletic performance. 

We analyzed the common variants of *MYRF*, *SOX10* and *OLIG2*, identified previously in the European population, in a large cohort of elite athletes. The obtained results were additionally verified using an independent validation group. Statistical analysis did not show any significant differences between the two groups in the alleles and genotypes frequency. Even though the athletes did not differ from the non-athlete controls in the distribution of most SNPs analyzed, some interesting differences from a random distribution of several variants were noted. Thus, in the validation group, two SNPs showed a correlation with the level of sport achievement among the athletes. Minor alleles of rs7943728 (intronic variant) and rs61747222 (missense variant, p.Arg311His) were associated with higher sport performance. Allele A rs7943728, relatively common in the European non-Finnish population (MAF = 0.19) was underrepresented in the high-elite group. In contrast, allele A rs61747222 variant rare in the non-Finnish European group (MAF = 0.03) but relatively frequent in the Finnish population (MAF = 0.11) was over-represented among the high-elite athletes. 

The functional significance of the variants associated with sport performance is not clear. According to the GTEx Portal database https://www.gtexportal.org (accessed on 13 June 2020) the *MYRF* gene expression is higher for both minor alleles. In turn, the missense rs61747222 variant (p.Arg311His) is predicted (SIFT, PolyPhen, CADD) not to affect protein function, in accordance with the rather conservative nature of this substitution. However, some fine differences between the two isoforms of the MYRF transcription factor, subtly affecting the expression of its client genes, cannot be excluded. This would explain the contrasting relations of the two minor variants, rs61747222 and rs7943728 with the highest level of sports performance despite both enhancing MYRF expression.

One of the polymorphisms discussed above has already been highlighted by GWAS data https://www.ebi.ac.uk/gwas/studies/GCST005650 (accessed on 9 July 2020). A strong association of allele A of rs7943728 was found with the level of serum metabolites of carnitine in chronic kidney disease. Carnitine is required for the import of long-chain fatty acids to mitochondria and therefore is critical for oxidation and consequently for energy production. This is particularly important in endurance sports when the carbohydrate reserves of the body are insufficient to support the energy requirements. If the rs7943728 polymorphism also affect carnitine metabolism in healthy participants, its association with athletic performance would be explained by its predicted influence on energy metabolism.

Differences in the efficiency of neural circuits between endurance and combat athletes could be expected due to the need for precision of repetitive movement in the former vs. high response speed in the latter. On the other hand, the high efficiency of all systems of the human body is indispensable for high-level sport competition. 

Summarizing, the presented results indicate that SNPs of *MYRF* (s174528, rs2238001, rs198459, rs61747222) and *SOX10* (rs139884, rs149435516) could be genetic factors weakly predisposing for successful athletic performance.

## Figures and Tables

**Table 1 genes-12-00262-t001:** Study group structure.

Status	*n*	Age at Enrollment (Mean ± SD, Years)	% Females	High Elite (*n*)	Elite (*n*)	Non Elite (*n*)
Elite athletes	642	22.48 ± 4.8	28.7	159	220	263
Endurance sports	221	23.43 ± 4.8	37.6	73	98	50
Power sports	186	23.62 ± 4.8	33.9	38	80	68
Combat sports	235	20.64 ± 4.4	16.2	48	42	145
Controls	724	22.72 ± 2.70	48.2	n/a	n/a	n/a
Total	1597					

**Table 2 genes-12-00262-t002:** Validation group structure.

Study Groups	*n*	Age at Enrollment (Mean ± SD, Years)	% Females	High Elite (*n*)	Elite (*n*)	Non Elite (*n*)
Elite athletes:	257	43.3 ± 13.0	23.74	78	81	98
Endurance sports	102	46.4 ± 12.1	20.59	28	33	41
Power sports	82	49.2 ± 15.2	31.71	26	24	32
Combat sports	73	32.3 ± 11.4	20.55	24	24	25
Control subjects	281	27.1 ± 9.5	36.30	n/a	n/a	n/a
Total	538					

**Table 3 genes-12-00262-t003:** Analyzed single nucleotide polymorphisms.

Genes	Polymorphism
*MYRF* (11q12.3)	rs2286008 rs2238001 ^2^ rs198459 ^2^ rs200370195 rs139124174 rs149803 ^2^ rs139799827 rs143193141 ^1^ rs143144043 rs143799782 rs146348968 rs139188067 rs143059056 rs174528 ^2^ rs144177087 rs141597490 rs7943728 ^2^ rs201384645 rs61747222 ^2^ rs35113793
*SOX10* (22q13.1)	rs148688873 rs149435516 ^2^ rs147334218 rs139884 ^2^ rs147817756 rs142113652 ^1^ rs138500876
*OLIG2* (21q22.11)	rs762178

^1^ SNPs excluded from the analysis due to low call rate of the genotyping assay. ^2^ SNPs analyzed in validation group.

## Data Availability

All data supporting results are avaible in Appendix A.

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
