# Peer review of "Common Myelin Regulatory Factor Gene Variants Predisposing to Excellence in Sports"

_genes, 2021, doi:10.3390/genes12020262_

Round 1
Reviewer 1 Report
Thank you for this interesting and alternative look at SNPs and athletic performance, refreshing to look at this view rather than energy systems and muscle type genes. I understand that this journal is concentrating on the genetic aspect of this project however I feel there needs to be a limitations and future study aspect that includes the following: 1.The differences between sports, these are broken down into primarily physical attribute categories such as endurance and power, however the types of sports that these genes may have more influence over should be discussed, sports such as soccer, basketball and tennis require different forms of agility, co-ordination, proprioception, player awareness compared with say 1500m running or marathon running. Not to say that one requires more or less motor skill but they are different and may be affected more from other aspects and these all differ from primarily skill sports such as archery. 2. Explaining where the research could go next? A more homogenous group of one particular sport but breaking this up into its differing levels is an example of where you can take out some variables of the sport demands. I.e. soccer players from amateur to elite professional, are these particular SNPs allowing some players to reach higher levels? if so at one point does a gene variant prevent a players from reaching higher levels in their sports. 3. In the authors opinion should these genes be studied for athletic performance as vigorously as genes such as ACE and ACTN3? 4. Explaining the limitations of such a study when not considering other genetic variables that may affect energy systems, muscle fibre etc.Author Response
Thank you very much for revision and comments.
1 Yes, indeed it is very difficult to compare motor skills between different sports. In our study we omitted samples from athletes competing in field as archery or teams sport (as generally problematic to assign to power or endurance subgroup). So we think that our subgroups are relatively homozygous if you consider motor skills requirements. At the same point we hypothesized that athletes achieving national or world class levels inside any subgroup should be “physically” comparable to each other so the other factors then directly connected with muscle strength or endurance become significant.
2 Since we realize that there is some association between relatively common variant in MYRF and sport performance it is as you suggest good idea to further investigate in such variant or even more whole MYRF and SOX10 etc genes to looking for rare variant in more homozygous group we were thinking about maybe marathon runners which is massive movement in Poland – then compare amateur vs. professionals – but such study also have limitation due the limited number of professional in one sport field. There were some study in musician and juggling people proving that such activates can influence oligodendrocyte production. But it can also be opposite efficient myelination is necessary for excellence in such activities. Another idea can be study association in check players where intuitively can has more influence then another sport related genes. We can imagine that one variant can help achieve higher level but not without genetic background but at the same time it is hard to image that one variant that allowed to become a professional athlete but can prevent from reaching higher levels – as it is polygenetic dependent so impact of such variant should be compensated by other mechanisms. It seems that the influence of a single variation could be strong enough to prevent professional training at all but not progress in training.
3 The athletic performance for sure is dependent from whole genetic background but of course there are genes (variants) which influence is putatively more pronounced within the aspect of athletic performance under study.. For instance ACTN3 and ACE are genes which have a recognized impact on athletic perfomance. Other genes have more problematic influence to include them into our study. We hypothesize that efficiency of nervous system is also essential but it is much more difficult to analyze using genetic association methods. And from that point of view considering progress of science and trouble to find new obvious association it is worth to study in sport context.
4 Genes (variants) directly associated with muscle power and endurance are obvious and very often of basis of the real success in sport, as the elite level athletes have relatively the same level of organism efficacy. Then the minute difference can play role but as the athletes performance is a polygenetic trait each variant has a minute impact. Additionally, each individual can bear different variants influencing the same sport-related phenotype. All in all, genetic association studies are always dependent on the sample size, as well as the rarity of variant studied.
Reviewer 2 Report
In this study authors sought to investigate gene variants predisposing to elite performance in sports in relation to myelination. Findings were that genetic traits in athletes were not different from those in the control group, although some variants seemed to be weakly associated with the level of performance. I compliment the authors for the quality of the data proposed and the clarity of the manuscript – succinct and straight to the point. Organization and flow of the paper as well as the interpretation of findings are excellent.
I have a few minor remarks that the authors are suggested to consider.
Line 87: I would not say “influence”. I would rather say “predispose” or “facilitate”.
Line 91-92/ 97-98: it is not clear how the groups were recruited. I think more info is needed here for the reader to understand the exact level of competition of the athletes. I think this is key in order to verify that there was a clear difference between the athletes and the control group. In this context, could this have been a confounding factor? If so, this should be acknowledged in discussion.
Line 267: strictly speaking, energy cannot be produced. I think it is more appropriate to say “energy metabolism”.
Line 271-272: I think that either the authors expand on these trends or they avoid mentioning them. As it is, this part is too vague and may be misleading.
Author Response
Thank you very much for revision and comments. We corrected the minor pointed out phrases.
Line 91-98
The participants of study were recruited independently out of two research centers (Józef Piłsudski University of Physical Education and Gdansk University of Physical Education and Sport). Samples from athletes were collected from active usually during national competitions or sports camps. Control group were sedentary participants declared themselves as not doing sport usually recruited from two mentioned above research centers. The detailed inclusion criteria were in the cited manuscript Peplonska et al 2019 and we just did not repeat it.
Line 271-272
After consideration we think that it is true that this sentence is unnecessary in this place